# Numerical Simulation and Design Recommendations for Web Crippling Strength of Cold-Formed Steel Channels with Web Holes under Interior-One-Flange Loading at Elevated Temperatures

**Zhiyuan Fang** [1], **Krishanu Roy** [1,2,*], **Hao Liang** [1], **Keerthan Poologanathan** [3], **Kushal Ghosh** [4], **Abdeliazim Mustafa Mohamed** [5,6] **and James B. P. Lim** [1,2]

1   Department of Civil and Environmental Engineering, The University of Auckland,
    Auckland 1023, New Zealand; zfan995@aucklanduni.ac.nz (Z.F.); hlia929@aucklanduni.ac.nz (H.L.);
    james.lim@auckland.ac.nz (J.B.P.L.)
2   School of Engineering, The University of Waikato, Hamilton 3216, New Zealand
3   Faculty of Engineering and Environment, University of Northumbria, Newcastle upon Tyne NE7 7YT, UK;
    keerthan.poologanathan@northumbria.ac.uk
4   Department of Civil Engineering, National Institute of Technology, South Sikkim 737139, Sikkim, India;
    kushal@nitsikkim.ac.in
5   Department of Civil Engineering, College of Engineering, Prince Sattam Bin Abdulaziz University,
    Al-Kharj 16278, Saudi Arabia; a.bilal@psau.edu.sa
6   Building & Construction Technology Department, Bayan University, Khartoum 210, Sudan
*   Correspondence: krishanu.roy@auckland.ac.nz or krishanu.roy@waikato.ac.nz

**Abstract:** This paper investigates the interior-one-flange web crippling strength of cold-formed steel channels at elevated temperatures. The stress-strain curves of G250 and G450 grade cold-formed steel (CFS) channels at ambient and elevated temperatures were taken from the literature and the temperatures were varied from 20 to 700 °C. A detailed parametric analysis comprising 3474 validated finite element models was undertaken to investigate the effects of web holes and bearing length on the web crippling behavior of these channels at elevated temperatures. From the parametric study results, it was found that the web crippling strength reduction factor is sensitive to the changes of the hole size, hole location, and the bearing length, with the parameters of hole size and hole location having the largest effect on the web crippling reduction factor. However, the web crippling strength reduction factor remains stable when the temperature is changed from 20 to 700 °C. Based on the parametric analysis results, the web crippling strength reduction factors for both ambient and elevated temperatures are proposed, which outperformed the equations available in the literature and in the design guidelines of American standard (AISI S100-16) and Australian/New Zealand standard (AS/NZS 4600:2018) for ambient temperatures. Then, a reliability analysis was conducted, the results of which showed that the proposed design equations could closely predict the reduced web crippling strength of CFS channel sections under interior-one-flange loading conditions at elevated temperatures.

**Keywords:** web crippling; proposed equations; elevated temperatures; interior-one-flange loading; web hole; finite element analysis; cold-formed steel

## 1. Introduction

In recent years, the popularity of cold-formed steel (CFS) has increased in the construction industry, due to their numerous advantages, such as superior strength to weight ratio, stiffness, and ease of construction, when compared to hot-rolled steel structures [1–3]. The applications of these CFS members (particularly CFS channels) often include beams [4–7], columns [8–16], shear walls [17], and cladding systems [18]. However, localized web failure can occur near the web holes of CFS channels, especially under transverse concentrated loads. The fire safety of these CFS channels is also essential to minimize the damage caused

by fire-related accidents [2,19]. This paper intends to investigate the effect of fire loading on the interior-one-flange (IOF) web crippling strength of CFS channels with web holes.

Extensive research works are available in the literature on the web crippling strength of CFS channels at ambient temperatures [20–33]. However, very limited research studies are available in the literature for the IOF web crippling capacity of these CFS channels at elevated temperatures. In addition, no information is available in the current design standards of CFS, explaining how the effect of fire loading can reduce the web crippling capacity of CFS channels from ambient to elevated temperatures. The lack of design information makes it difficult for practicing engineers and researchers to predict the web crippling capacity of CFS channels subjected to one-flange loading at elevated temperatures.

Recent studies have started to focus on the material behavior of CFS sections at elevated temperatures. Imran et al. [34] recently proposed numerical equations to evaluate the strength reduction factors of square, rectangular, and circular CFS hollow sections at elevated temperatures. Coupons were cut from these hollow sections and loaded under temperatures ranging from 20 to 800 °C. The main aim of their study was to determine the reduction in material properties. Furthermore, Kankanamge and Mahendran [35] proposed the updated equations for reduction factors of the stress-strain relationship for both the normal and high strength steels of different grades at elevated temperatures. A similar study was completed by Ranawaka and Mahendran [36], who proposed empirical equations for determining the stress-strain relationship of both the normal and high strength steels at elevated temperatures. Chen and Young [37] reported mechanical property data for G550 and G450 grades of CFS sections by conducting tensile coupon tests under both the steady and transient temperature conditions. Lim and Young [38] used the stress-strain relationships of Chen and Young [37] to determine the effect of fire loading on the capacity of CFS bolted connections.

Alongside the studies reported in the literature on reduced mechanical properties of CFS sections at elevated temperatures, some researchers also focused on the structural behavior of different CFS sections at elevated temperatures and that are subject to different loading conditions. Multiple investigations have been completed to determine the effect of elevated temperatures on CFS beams. Landesmann and Camotim [39] presented a FE investigation on the distortional buckling behavior of CFS single-span lipped channel beams under elevated temperatures. Laim et al. [40] completed a study to understand the structural behavior of CFS beams in fire. Kankanamge and Mahendran [41] presented a validated FE model to determine the structural behavior of CFS lipped channel beams under bending at elevated temperatures.

The structural behavior of CFS columns at elevated temperatures has been studied by researchers to date. Gunalan et al. [31] carried out the experimental and numerical investigation on the local buckling behavior of CFS lipped and unlipped channel columns under simulated fire loading. Gunalan et al. [42] also presented a study on the flexural-torsional buckling interaction of CFS lipped channel columns at ambient and elevated temperatures. Ranawaka and Mahendran [43] conducted a study to determine the distortional buckling behavior of CFS lipped channel columns at elevated temperatures. Chen and Young [37] performed a numerical study to understand the behavior of CFS lipped channel columns at elevated temperatures. Feng and Wang [44] investigated the axial strength of CFS channel columns under ambient and elevated temperatures.

Of note, most of the research studies available in the current literature focus on the behavior of CFS sections under compression and torsional loading at elevated temperatures. No research is available in the literature that investigated the effects of web holes on the web crippling strength of CFS channels at elevated temperatures. Furthermore, the current design specifications, such as ASCE [45], EN 1993-1-1 [46], and BS 5950 [47] do not provide any guidelines for CFS channels with web holes at elevated temperatures. However, AISI [48] and AS/NZ:4600 [49] offer reduction factor equations for CFS channels with web holes under IOF and end-one-flange (EOF). However, these are focused on channels with web holes that offset to the bearing edge and are applicable only at ambient temperatures.

Lian et al. [29,30] proposed strength reduction factors for determining the reduced web crippling capacity of CFS channels with web holes subjected to IOF loading. However, the reduction factors of Lian et al. [29,30] are only applicable at ambient temperatures and do not cover the case of elevated temperatures. This issue is addressed in the current research.

This paper investigates the feasibility of using the same reduction factor equations of Lian et al. [29,30] from ambient temperatures to elevated temperatures. Figure 1 shows the symbol definitions used for the dimensions of CFS channels considered in this study. Based on the results of 3474 finite element (FE) models, the parametric effects of web holes and bearing length on the web crippling strength of CFS channels were investigated. From the parametric analysis results, design recommendations are proposed for the reduced IOF web crippling strength of CFS channels at elevated temperatures. Then, a comparison of results from the proposed equations and the equations of Lian et al. [29,30] and AISI [48] and AS/NZS [49] was made and showed that the proposed equations outperformed the others. Next, a reliability analysis was conducted, which showed that the proposed equations could closely predict the reduced web crippling strength of CFS channels when loaded with IOF loading at elevated temperatures.

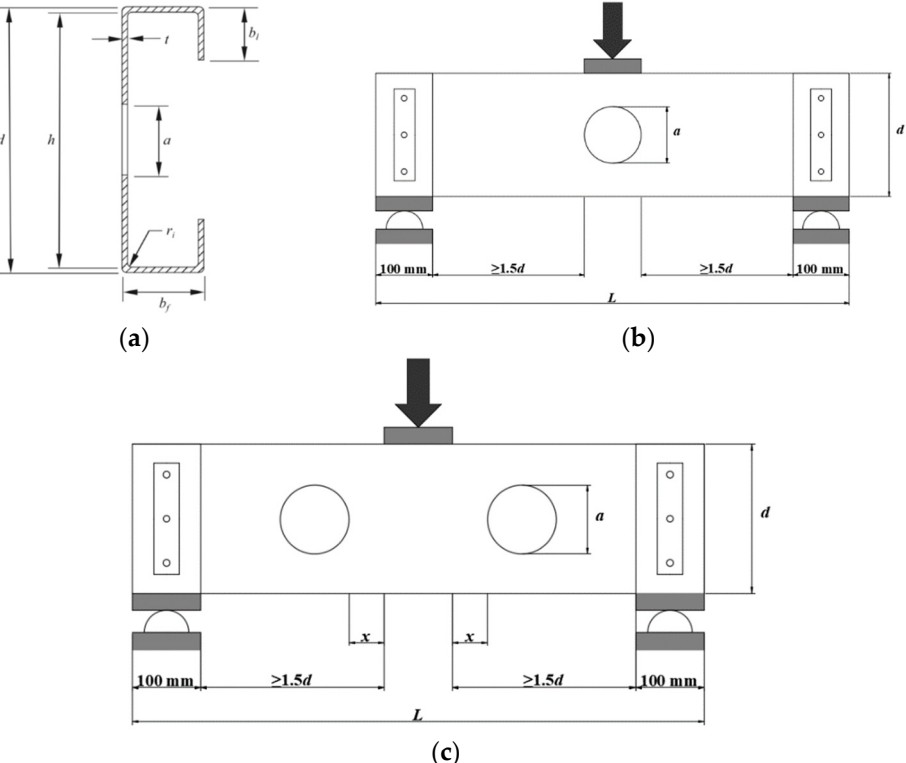

**Figure 1.** Definition of symbols and loading cases: (**a**) Section dimensions; (**b**) sections with centered holes from the bearing plate under IOF; (**c**) sections with offset holes from the bearing plate under IOF.

## 2. Summary of the Experimental Investigation

A total of 61 experimental test results of CFS channels with web holes subjected to IOF loading were reported in Lian et al. [29,30]. The cases of both fastened flange and unfastened flange are considered in the experimental tests. In addition, the hole of the specimens was located as centered beneath the bearing plate or with a horizontal clear distance to the near edge of the bearing plate. The experimental results matched well with the validated FE models in terms of failure modes and failure loads.

## 3. Numerical simulation

### 3.1. Development of the Finite Element Model

A nonlinear elasto-plastic FE model was developed using a finite element analysis (FEA) software named ABAQUS [50] to simulate the IOF web crippling behavior of CFS channels with web holes (see Figure 2). The CFS channels were modelled using S4R shell elements with a mesh size of 5 mm× 5 mm. In total, around 3000 elements were used. The upper endplate was modelled using rigid quadrilateral shell elements (R3D4) with a mesh size of 10 mm× 10 mm. In total, 350 elements were used to model the upper endplate. Figure 3 illustrates a typical FE mesh.

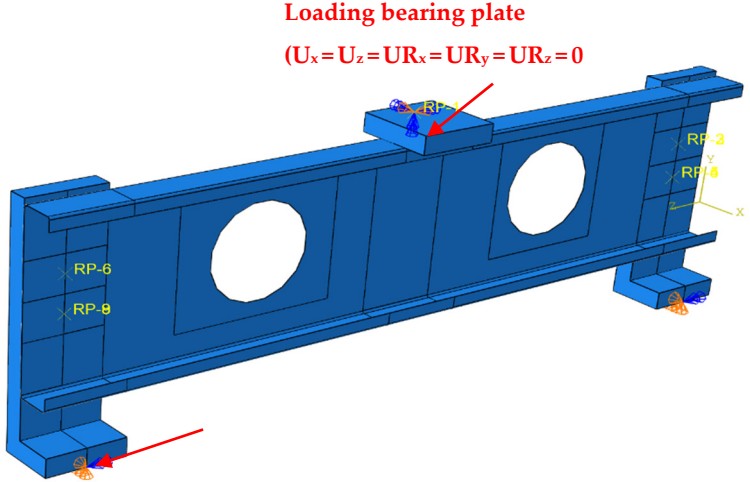

**Figure 2.** Boundary conditions used in FE models.

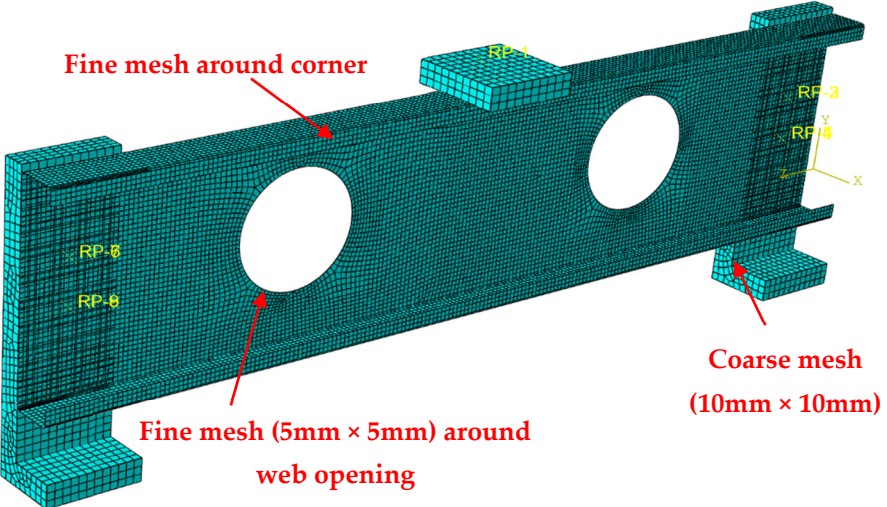

**Figure 3.** FE meshing types.

The stress-strain curves of 1.55 and 1.95 mm thick G250 CFS and 1.50 and 1.90 mm G450 steels at elevated and ambient temperatures were taken from Kankanamge and Mahendran [35] and used in the FE model.

The interface of bearing plate and channel section was modelled using the surface-to-surface contact option. The target surface was the bearing plate, whereas the contact surface was selected to be the channel section. No penetration of the two contact surfaces was permitted. Displacement control was applied to model the vertical load applied to the channels through the reference node of the top bearing plate. A similar modelling method was employed by Fang et al. [16], Chen et al. [51–53], and Roy et al. [54–56].

### 3.2. FE Validation

A total of 61 experimental test results of Lian et al. [29,30] were used to validate the FE model. As shown in Tables 1 and 2, the average ratios of experimental to FEA strengths ($P_{EXP}/P_{FEA}$) are 1.00 and 0.94, respectively for the CFS channels with unfastened and fastened flanges at ambient temperatures. Therefore, the FE models could closely predict the IOF web crippling strength of perforated CFS channels at ambient temperatures.

**Table 1.** Comparison of experimental results with FEA results for sections with unfastened flanges.

| Specimen ID | Web | Flange | Lip | Bend Radius | Thickness | Hole Dia. | Bearing Length | Yield Stress | Exp. Load | FEA Result | $P_{EXP}/P_{FEA}$ |
|---|---|---|---|---|---|---|---|---|---|---|---|
| | $d$ | $b_f$ | $b_l$ | $r$ | $t$ | $a$ | $N$ | $f_y$ | $P_{EXP}$ | $P_{FEA}$ | |
| | (mm) | (mm) | (mm) | (mm) | (mm) | (mm) | (mm) | (MPa) | (kN) | (kN) | |
| 1 | 141.82 | 60.63 | 13.66 | 4.8 | 1.27 | 0.00 | 100 | 639.8 | 10.78 | 11.21 | 0.96 |
| 2 | 142.27 | 60.41 | 13.86 | 4.8 | 1.27 | 83.66 | 100 | 639.8 | 10.17 | 10.66 | 0.95 |
| 3 | 142.31 | 59.94 | 13.97 | 4.8 | 1.28 | 83.64 | 100 | 639.8 | 10.32 | 10.84 | 0.95 |
| 4 | 142.24 | 60.37 | 13.9 | 4.8 | 1.27 | 0.00 | 120 | 639.8 | 11.64 | 12.05 | 0.97 |
| 5 | 142.11 | 60.2 | 13.97 | 4.8 | 1.28 | 83.68 | 120 | 639.8 | 10.54 | 11.32 | 0.93 |
| 6 | 142.42 | 60.2 | 13.6 | 4.8 | 1.27 | 83.73 | 120 | 639.8 | 10.57 | 10.82 | 0.98 |
| 7 | 142.4 | 59.79 | 13.28 | 4.8 | 1.28 | 0.00 | 150 | 639.8 | 12.60 | 12.93 | 0.97 |
| 8 | 142.17 | 59.88 | 12.95 | 4.8 | 1.28 | 55.04 | 150 | 639.8 | 12.49 | 12.52 | 1.00 |
| 9 | 142.37 | 60.26 | 13.22 | 4.8 | 1.28 | 54.66 | 150 | 639.8 | 12.51 | 12.45 | 1.01 |
| 10 | 202.04 | 64.79 | 14.78 | 5 | 1.39 | 0.00 | 100 | 649.6 | 12.15 | 11.92 | 1.02 |
| 11 | 202.03 | 64.86 | 14.98 | 5 | 1.39 | 79.25 | 100 | 649.6 | 11.70 | 11.47 | 1.02 |
| 12 | 202.07 | 65.01 | 14.95 | 5 | 1.39 | 79.26 | 100 | 649.6 | 11.59 | 12.18 | 0.95 |
| 13 | 202.11 | 65.45 | 14.39 | 5 | 1.39 | 119.07 | 100 | 649.6 | 10.81 | 10.48 | 1.03 |
| 14 | 202 | 65 | 14.73 | 5 | 1.39 | 0.00 | 120 | 649.6 | 12.98 | 12.62 | 1.03 |
| 15 | 202 | 65.04 | 14.82 | 5 | 1.39 | 79.31 | 120 | 649.6 | 11.63 | 11.99 | 0.97 |
| 16 | 202.66 | 65.35 | 14.57 | 5 | 1.38 | 119.22 | 120 | 649.6 | 11.16 | 10.90 | 1.02 |
| 17 | 202 | 65.06 | 14.88 | 5 | 1.39 | 79.32 | 120 | 649.6 | 12.21 | 12.41 | 0.98 |
| 18 | 202.26 | 65.39 | 14.5 | 5 | 1.39 | 119.39 | 120 | 649.6 | 10.95 | 10.39 | 1.05 |
| 19 | 202.01 | 65.04 | 14.98 | 5 | 1.45 | 0.00 | 150 | 649.6 | 14.51 | 14.40 | 1.01 |
| 20 | 202.01 | 64.96 | 15.02 | 5 | 1.43 | 79.35 | 150 | 649.6 | 12.98 | 13.27 | 0.98 |
| 21 | 202 | 65.09 | 15 | 5 | 1.39 | 79.32 | 150 | 649.6 | 13.23 | 12.38 | 1.07 |
| 22 | 303.18 | 87.91 | 18.83 | 5 | 1.98 | 0.00 | 100 | 670.6 | 24.57 | 24.74 | 0.99 |
| 23 | 302.58 | 88.61 | 19.28 | 5 | 2.06 | 178.89 | 100 | 670.6 | 21.89 | 24.08 | 0.91 |
| 24 | 303.05 | 88.2 | 18.99 | 5 | 1.98 | 179.00 | 100 | 670.6 | 22.85 | 21.13 | 1.08 |
| 25 | 303.07 | 87.95 | 18.26 | 5 | 1.96 | 0.00 | 120 | 670.6 | 25.16 | 25.42 | 0.99 |
| 26 | 303.05 | 88.03 | 18.32 | 5 | 2.06 | 178.99 | 120 | 670.6 | 23.24 | 24.04 | 0.97 |
| 27 | 303.03 | 87.99 | 18.3 | 5 | 1.98 | 179.00 | 120 | 670.6 | 23.29 | 21.47 | 1.08 |
| 28 | 303.03 | 88.54 | 18.97 | 5 | 1.99 | 0.00 | 150 | 670.6 | 28.24 | 27.36 | 1.03 |
| 29 | 302.9 | 88.47 | 19.03 | 5 | 2.06 | 178.55 | 150 | 670.6 | 24.40 | 23.37 | 1.04 |
| 30 | 303.63 | 88.25 | 19.11 | 5 | 1.99 | 178.66 | 150 | 670.6 | 24.18 | 21.88 | 1.11 |
| Average | | | | | | | | | | | 1.00 |
| COV | | | | | | | | | | | 0.05 |

**Table 2.** Comparison of experimental results with FEA results for sections with fastened flanges.

| Specimen ID | Web | Flange | Lip | Bend Radius | Thickness | Hole Dia. | Bearing Length | Yield Stress | Exp. Load | FEA Result | $P_{EXP}/P_{FEA}$ |
|---|---|---|---|---|---|---|---|---|---|---|---|
| | $d$ | $b_f$ | $b_l$ | $r$ | $t$ | $a$ | $N$ | $f_y$ | $P_{EXP}$ | $P_{FEA}$ | |
| | (mm) | (mm) | (mm) | (mm) | (mm) | (mm) | (mm) | (MPa) | (kN) | (kN) | |
| 1 | 142.49 | 60.33 | 13.79 | 4.8 | 1.29 | 0.00 | 100 | 639.8 | 11.14 | 12.21 | 0.91 |
| 2 | 142.56 | 60.11 | 13.78 | 4.8 | 1.29 | 84.67 | 100 | 639.8 | 10.89 | 11.85 | 0.92 |
| 3 | 142.48 | 60.06 | 13.7 | 4.8 | 1.29 | 83.59 | 100 | 639.8 | 10.97 | 11.39 | 0.96 |
| 4 | 142.38 | 60.21 | 13.68 | 4.8 | 1.29 | 0.00 | 120 | 639.8 | 12.33 | 13.13 | 0.94 |
| 5 | 142.26 | 60.22 | 13.67 | 4.8 | 1.29 | 83.78 | 120 | 639.8 | 11.97 | 12.53 | 0.96 |

**Table 2.** *Cont.*

| Specimen ID | Web | Flange | Lip | Bend Radius | Thickness | Hole Dia. | Bearing Length | Yield Stress | Exp. Load | FEA Result | $P_{EXP}/P_{FEA}$ |
|---|---|---|---|---|---|---|---|---|---|---|---|
| | $d$ | $b_f$ | $b_l$ | $r$ | $t$ | $a$ | $N$ | $f_y$ | $P_{EXP}$ | $P_{FEA}$ | |
| | (mm) | (mm) | (mm) | (mm) | (mm) | (mm) | (mm) | (MPa) | (kN) | (kN) | |
| 6 | 142.53 | 60.29 | 13.91 | 4.8 | 1.29 | 83.77 | 120 | 639.8 | 11.69 | 11.83 | 0.99 |
| 7 | 142.18 | 60.12 | 13.19 | 4.8 | 1.28 | 0.00 | 150 | 639.8 | 13.48 | 14.11 | 0.96 |
| 8 | 142.35 | 60.07 | 13.2 | 4.8 | 1.28 | 55.26 | 150 | 639.8 | 13.04 | 13.88 | 0.94 |
| 9 | 142.42 | 60.07 | 13.13 | 4.8 | 1.28 | 55.20 | 150 | 639.8 | 13.28 | 13.72 | 0.97 |
| 10 | 201.99 | 64.87 | 14.76 | 4.8 | 1.38 | 0.00 | 100 | 649.6 | 13.35 | 13.88 | 0.96 |
| 11 | 202.01 | 64.96 | 14.76 | 4.8 | 1.37 | 79.36 | 100 | 649.6 | 12.42 | 13.07 | 0.95 |
| 12 | 202.22 | 65.44 | 14.42 | 4.8 | 1.37 | 119.41 | 100 | 649.6 | 11.73 | 12.90 | 0.91 |
| 13 | 202.11 | 64.92 | 14.99 | 4.8 | 1.37 | 79.30 | 100 | 649.6 | 12.60 | 14.65 | 0.86 |
| 14 | 201.79 | 65.68 | 14.64 | 4.8 | 1.37 | 119.45 | 100 | 649.6 | 12.18 | 12.83 | 0.95 |
| 15 | 202.05 | 64.99 | 14.82 | 4.8 | 1.41 | 0.00 | 120 | 649.6 | 14.60 | 15.47 | 0.94 |
| 16 | 201.98 | 65.1 | 14.92 | 4.8 | 1.38 | 79.32 | 120 | 649.6 | 13.36 | 14.25 | 0.94 |
| 17 | 201.76 | 65.4 | 14.62 | 4.8 | 1.39 | 119.51 | 120 | 649.6 | 12.98 | 14.22 | 0.91 |
| 18 | 202 | 65.16 | 15.02 | 4.8 | 1.39 | 79.36 | 120 | 649.6 | 13.94 | 15.73 | 0.89 |
| 19 | 202.42 | 65.36 | 14.4 | 4.8 | 1.39 | 119.41 | 120 | 649.6 | 12.44 | 13.57 | 0.92 |
| 20 | 202 | 64.93 | 15 | 4.8 | 1.41 | 0.00 | 150 | 649.6 | 16.16 | 16.71 | 0.97 |
| 21 | 202.01 | 64.88 | 14.98 | 4.8 | 1.38 | 79.31 | 150 | 649.6 | 14.63 | 15.51 | 0.94 |
| 22 | 202.02 | 64.88 | 14.79 | 4.8 | 1.38 | 79.32 | 150 | 649.6 | 14.96 | 16.21 | 0.92 |
| 23 | 303.2 | 88.24 | 18.66 | 4.8 | 1.96 | 0.00 | 100 | 670.6 | 25.26 | 27.60 | 0.92 |
| 24 | 303.44 | 88.38 | 19.34 | 5 | 1.9 | 178.91 | 100 | 670.6 | 22.95 | 25.04 | 0.92 |
| 25 | 303.45 | 88.57 | 19.26 | 5 | 1.91 | 178.42 | 100 | 670.6 | 24.26 | 24.48 | 0.99 |
| 26 | 303.5 | 88.53 | 18.36 | 5 | 1.93 | 0.00 | 120 | 670.6 | 26.40 | 28.53 | 0.93 |
| 27 | 303.28 | 88.79 | 18.55 | 5 | 1.9 | 178.73 | 120 | 670.6 | 23.74 | 26.43 | 0.90 |
| 28 | 303.02 | 88.77 | 18.48 | 5 | 1.9 | 178.69 | 120 | 670.6 | 24.18 | 24.64 | 0.98 |
| 29 | 303.85 | 88.71 | 18.41 | 5 | 1.9 | 0.00 | 150 | 670.6 | 28.13 | 29.39 | 0.96 |
| 30 | 303.19 | 88.32 | 19.09 | 5 | 1.96 | 178.45 | 150 | 670.6 | 25.66 | 29.16 | 0.88 |
| 31 | 303.08 | 88.42 | 19.06 | 5 | 1.9 | 178.40 | 150 | 670.6 | 24.89 | 24.94 | 1.00 |
| Average | | | | | | | | | | | 0.94 |
| COV | | | | | | | | | | | 0.03 |

## 4. Current Design Rules

The calculation procedure for reduced IOF web crippling strength is available in Lian et al. [29,30] and AISI [48] and AS/NZS [49]. However, the procedure is applicable at ambient temperatures and does not necessarily work for elevated temperatures.

### 4.1. Current Design Standards

For IOF loading, where any portion of a web hole is not within the bearing length, the reduction factor, *R*, can be calculated using Equation (1) of AISI [48] and AS/NZS [49] as follows:

$$R = 0.90 - 0.047\frac{a}{h} + 0.053\frac{x}{h} \leq 1 \tag{1}$$

where, *a*, *h*, and *x* denote the hole diameter, depth of flat portion of the web, and the nearest distance between the web hole and the edge of bearing, respectively.

### 4.2. Reduction Factor Equations

Lian et al. [29,30] proposed the IOF web crippling strength reduction factor equations for CFS channels with web holes at ambient temperatures. These equations are limited to CFS channels with parametric ranges of $h/t \leq 157.8$, $N/t \leq 120.97$, $N/h \leq 1.15$, and $a/h \leq 0.8$. Moreover, these equations may not work for determining the web crippling strength of CFS channels at elevated temperatures.

The IOF web crippling strength reduction factor equations are proposed based on 3474 validated FE models. Unlike the equations proposed by Lian et al. [29,30], the parameter *N/h* is included in equations for both the cases of offset-hole and center-hole CFS channels.

## 5. Parametric Analysis

Using the validated FE model of CFS channels with web holes at ambient temperatures (described in Section 3 of this paper), an extensive parametric analysis was conducted to investigate the effects of fire loading on its web crippling strength. In total, 3474 FEA models were analyzed. It should be noted that the parametric analysis is mainly conducted to find out which parameters would lead to an unignorable change.

The selected failure modes of some sections are shown in Figures 4 and 5, and the plot of displacement-web crippling strength is shown in Figure 6. It was observed in Figures 4 and 5 that the out-of-plane deformation of the webs occurred gradually at the early stage of loading and continued to increase until failure occurred. The failure pattern was symmetrical, and failure occurred due to the formation of a local yield zone under the bearing plate. Moreover, the deformation due to the web crippling of channel sections at ambient temperatures was very low, when compared to the channel sections at elevated temperatures. This comparison shows that for the case of elevated temperatures, the web crippling resistance decreases considerably. Figure 6 shows a typical example of the load-deflection curve obtained from the FEA for the specimens with both unfastened and fastened flanges at ambient and elevated temperatures. As the load increases, the linear behavior was seen initially until reaching the yield point. The maximum stress occurred in the upper corner between the flange and the web of channels. Beyond the yield point, the plastic behavior began to spread through the channel section. When reaching the maximum load, the post-buckling strength of the channel section was achieved. In addition, it can be seen from Figure 6 that the web crippling strength decreased dramatically when the temperatures increased.

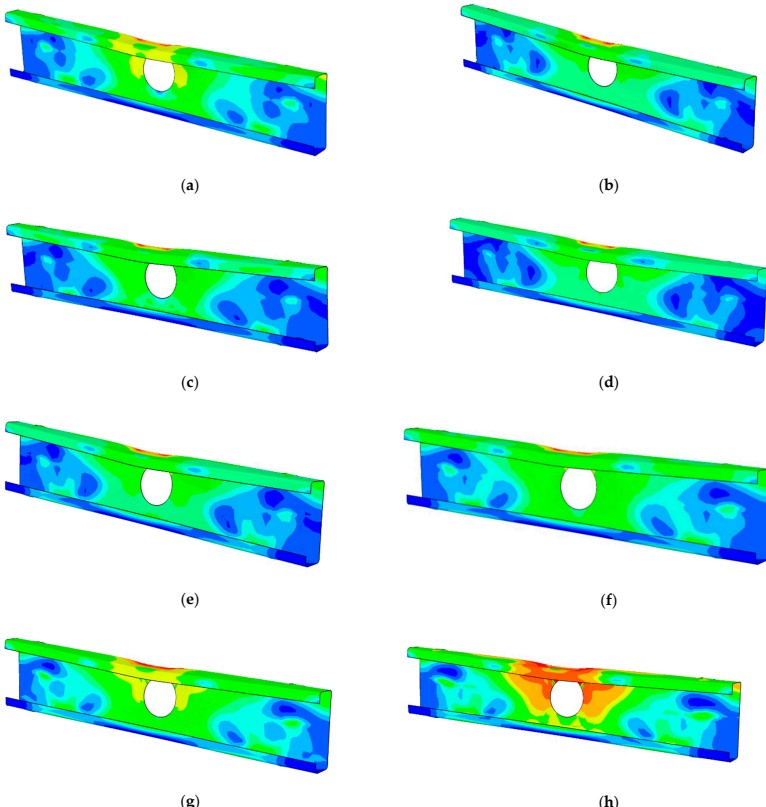

**Figure 4.** Failure modes of unfastened section (100 × 30 × 15-t1.55-N50-A0.6-FR) with the centered hole at different temperatures: (**a**) 20 °C; (**b**) 100 °C; (**c**) 200 °C; (**d**) 300 °C; (**e**) 400 °C; (**f**) 500 °C; (**g**) 600 °C; (**h**) 700 °C.

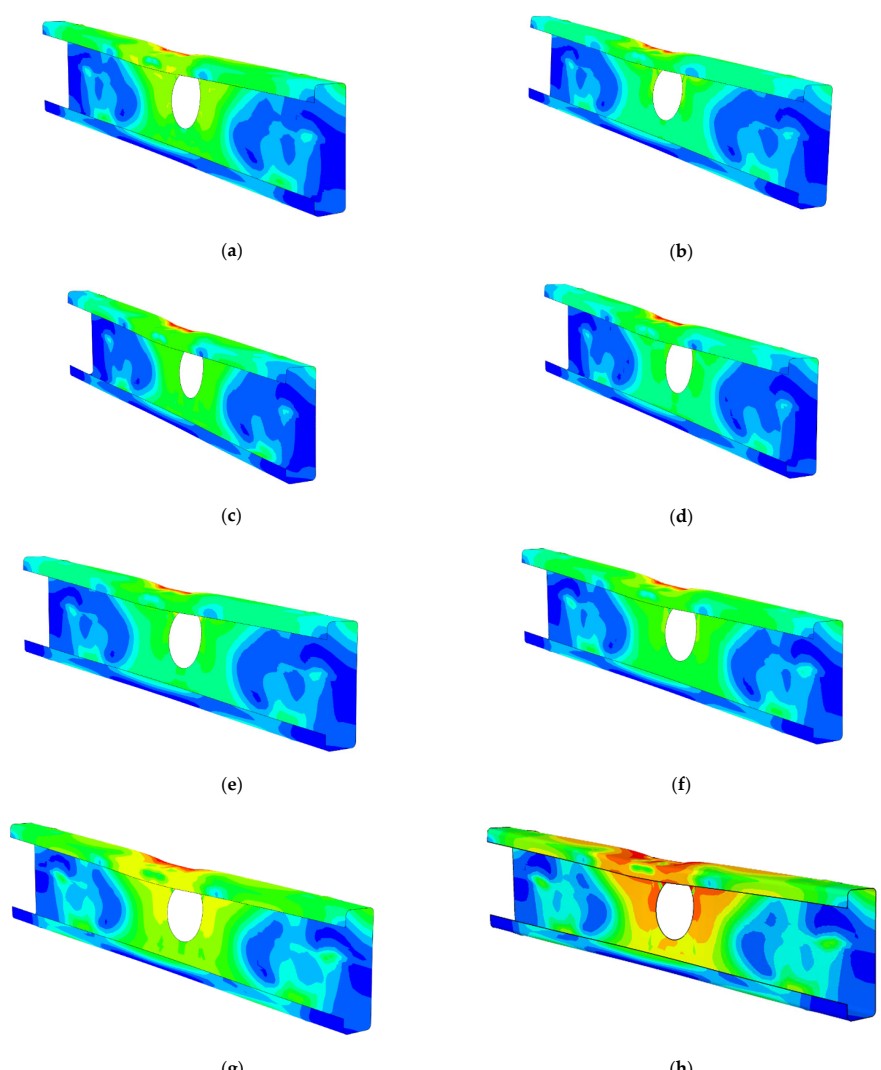

**Figure 5.** Failure modes of fastened section (100 × 30 × 10-t1.55-N50-A0.6-FX) with the centered hole at different temperatures: (**a**) 20 °C; (**b**) 100 °C; (**c**) 200 °C; (**d**) 300 °C; (**e**) 400 °C; (**f**) 500 °C; (**g**) 600 °C; (**h**) 700 °C.

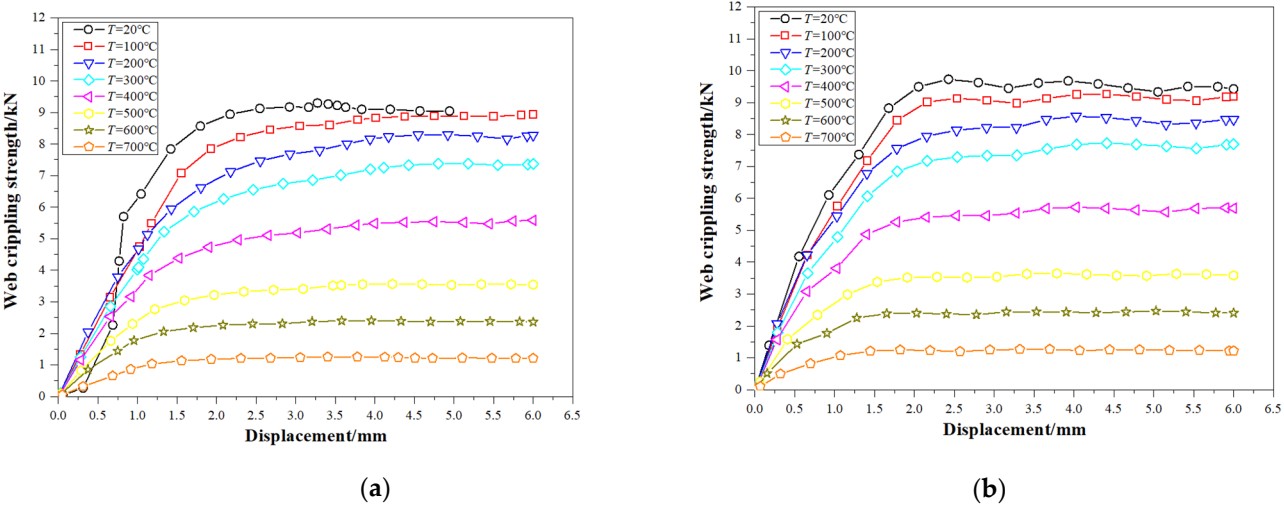

**Figure 6.** Load-displacement plot of selected sections under different temperatures for (**a**) unfastened section (100 × 30 × 15-t1.55-N50-A0.6-FR) with a centered hole and (**b**) fastened section (100 × 30 × 15-t1.55-N50-A0.6-FX) with a centered hole.

In addition, the detailed effects of each of these parameters on the web crippling strength of perforated CFS channels at elevated temperatures are discussed in the following sub-sections:

### 5.1. Effect of a/h, x/h and N/h Ratio

Figure 7 and Table 3 demonstrate the influence of the *a/h* ratio on the factor *R*. Figure 7 shows a decreasing trend in web crippling strength reduction factors as the *a/h* ratio increases from 0.2 to 0.8, with the change in the reduction factor as essentially identical for all temperatures groups. On the one hand, the factor *R* for offset-hole sections with unfastened and fastened flanges is identical, and the average factor *R* for these two sets of sections decreases from 0.97 to 0.81 and 0.96 to 0.92, respectively, when the ratio *a/h* rises from 0.2 to 0.8. The difference in factor *R* of centered-hole sections with unfastened and fastened flanges, on the other hand, is substantially higher. The average factor *R* for centered-hole sections with unfastened flanges reduced from 0.99 to 0.82, as indicated in Table 3. Meanwhile, when the *a/h* ratio was increased from 0.2 to 0.8, the factor *R* for CFS sections with fastened flanges reduced from 0.99 to 0.77.

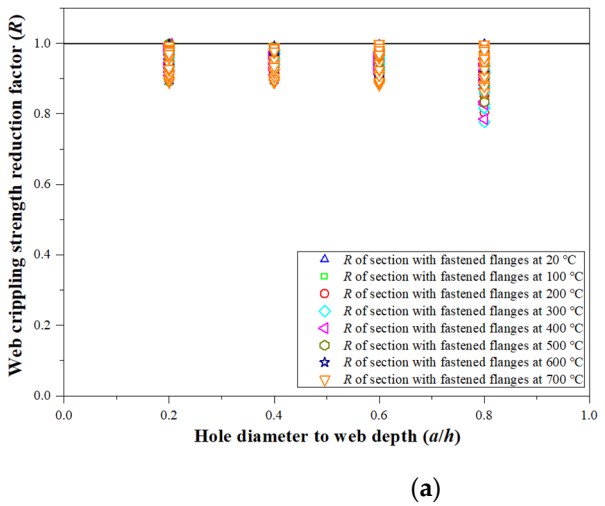

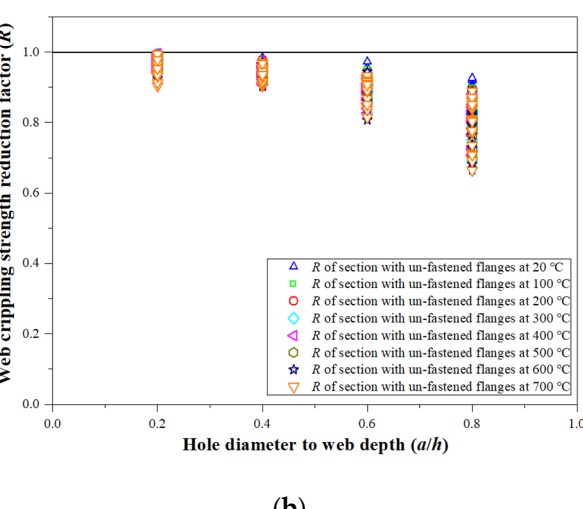

(**a**)　　　　　　　　　　　　　　　　　　　　　　　(**b**)

**Figure 7.** Web crippling strength reduction factor against *a/h* for cold-formed steel channel with (**a**) fastened flanges and offset web hole subjected to IOF and (**b**) unfastened flanges and offset web hole subjected to IOF.

**Table 3.** Average web crippling strength reduction factor (*R*) of investigated sections.

| Hole Position | | IOF Loading Condition | |
|---|---|---|---|
| | | Unfastened Flanges | Fastened Flanges |
| Centered hole | *a/h* = 0.2 | 0.99 | 0.99 |
| | *a/h* = 0.4 | 0.97 | 0.99 |
| | *a/h* = 0.6 | 0.93 | 0.94 |
| | *a/h* = 0.8 | 0.82 | 0.77 |
| Offset hole | *a/h* = 0.2 | 0.97 | 0.96 |
| | *a/h* = 0.4 | 0.94 | 0.96 |
| | *a/h* = 0.6 | 0.88 | 0.95 |
| | *a/h* = 0.8 | 0.81 | 0.92 |

The change in average factor *R* for offset-hole channels at varied temperatures remains steady between 0.85 and 0.99 as the *x/h* ratio is changed from 0.45 to 0.95.

The average factor *R* for sections with unfastened and fastened flanges decreased significantly by 4% and 5% on average for each set of *a/h* ratios, respectively, when the *N/h*

ratio was increased from 0.25 to 0.75. Figure 8 depicts the change in the factor $R$ as the $N/h$ ratio changes.

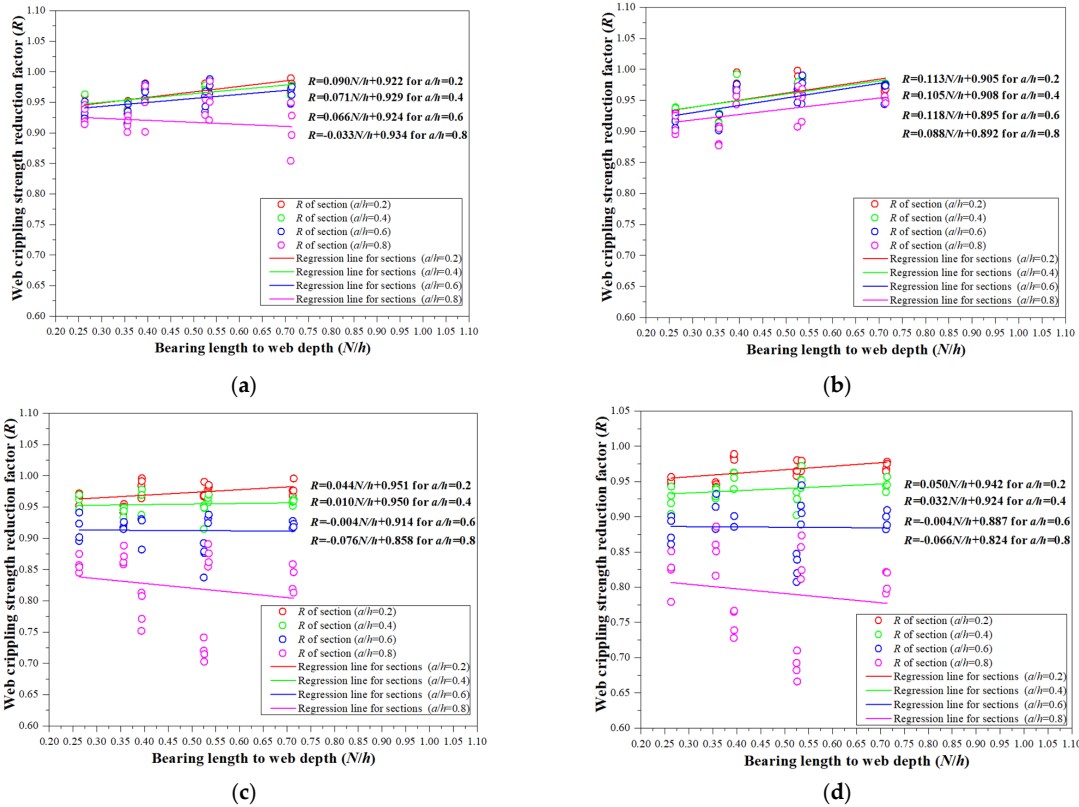

(a)     (b)

(c)     (d)

**Figure 8.** Web crippling strength reduction factor against $N/h$ for cold-formed steel channel with (**a**) fastened flanges and offset web hole subjected to IOF at 200 °C; (**b**) fastened flanges and offset web hole subjected to IOF at 600 °C; (**c**) unfastened flanges and offset web hole subjected to IOF at 200 °C; (**d**) unfastened flanges and offset web hole subjected to IOF at 600 °C.

### 5.2. Effect of Fastened Flanges

With varied $a/h$ ratios and hole positions, Figure 9 and Table 4 demonstrate the influence of fastened flanges on factor $R$. The average factor $R$ of CFS channels with fastened flanges is larger (by 5.3%) than those with unfastened flanges for sections with offset web holes. On the other hand, factor $R$ of centered-hole sections with fastened and unfastened flanges is fairly similar.

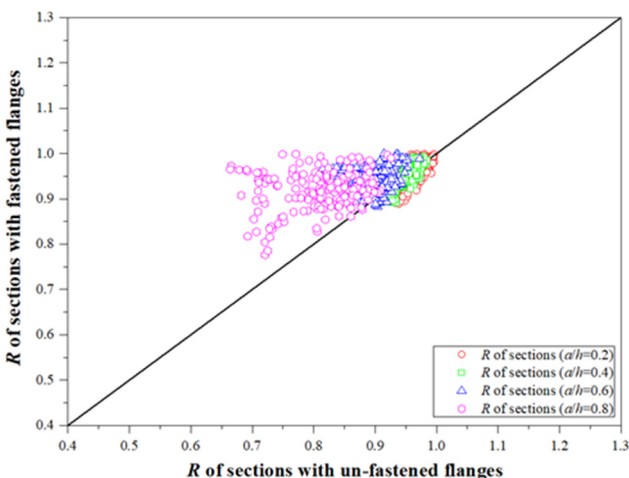

**Figure 9.** Web crippling strength reduction factor against unfastened/fastened flanges.

**Table 4.** Average web crippling strength reduction percentage (%) of investigated sections at elevated temperatures against ambient temperatures ($T = 20\ °C$).

| Temperatures | | IOF Loading Condition | |
|---|---|---|---|
| | | Unfastened Flanges | Fastened Flanges |
| Centered hole | T = 20 °C | - | - |
| | T = 100 °C | 5.67 | 5.01 |
| | T = 200 °C | 6.41 | 5.08 |
| | T = 300 °C | 19.52 | 16.19 |
| | T = 400 °C | 38.31 | 36.44 |
| | T = 500 °C | 61.28 | 60.67 |
| | T = 600 °C | 81.81 | 81.73 |
| | T = 700 °C | 89.57 | 89.67 |
| Offset hole | T = 20 °C | - | - |
| | T = 100 °C | 5.15 | 4.23 |
| | T = 200 °C | 4.88 | 3.42 |
| | T = 300 °C | 18.12 | 14.02 |
| | T = 400 °C | 37.25 | 35.33 |
| | T = 500 °C | 60.97 | 59.93 |
| | T = 600 °C | 81.94 | 81.45 |
| | T = 700 °C | 89.60 | 89.65 |

### 5.3. Effect of Elevated Temperatures

At elevated temperatures, the web crippling strength decrease percentages for CFS sections with unfastened flanges are slightly larger than those with fastened flanges. Figure 10 shows that when the temperatures rise from 20 to 700 °C, the web crippling strength ($P$) of CFS sections decreases. In the meantime, Table 4 indicates the average web crippling strength drop (from 5% to 90%) for each investigated temperatures group.

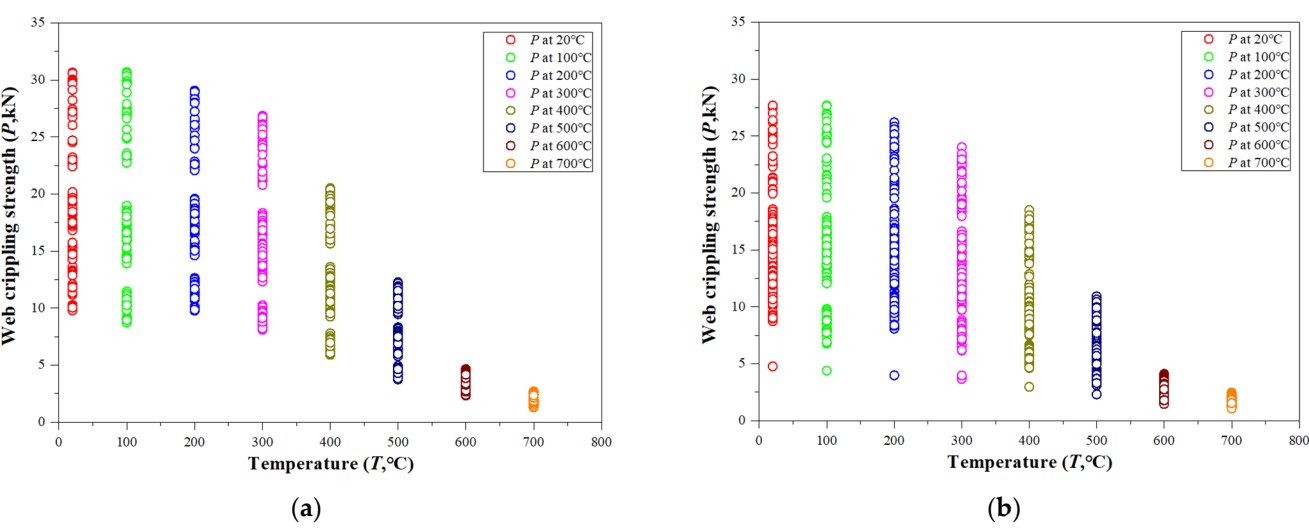

(**a**)        (**b**)

**Figure 10.** *Cont.*

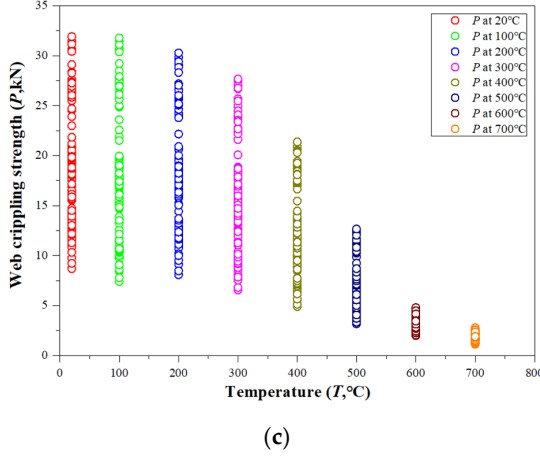
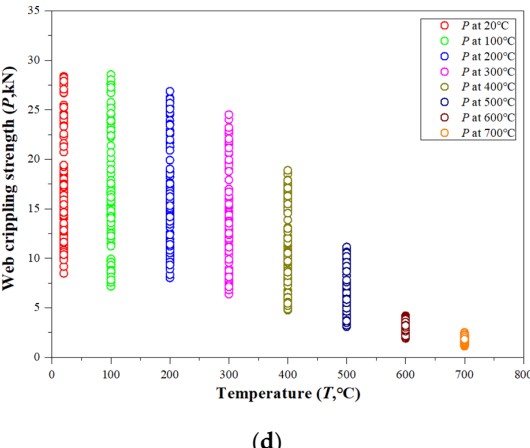

(**c**)                                                                                (**d**)

**Figure 10.** Web crippling strength against temperatures for cold-formed steel channel with (**a**) offset web hole and fastened flanges subjected to IOF; (**b**) offset web hole and unfastened flanges subjected to IOF; (**c**) centered web hole and fastened flanges subjected to IOF; (**d**) centered web hole and unfastened flanges subjected to IOF.

## 6. Proposed Design Equations and Reliability Analysis

The previous study showed that the FEA model was able to predict the web crippling strength of perforated CFS channels with more precision than the existing design guidelines. As a consequence of the FEA results, design equations in the form of web crippling strength reduction factors were proposed. The limits of the proposed equations are $h/t \leq 160$, $N/t \leq 120$, $N/h \leq 0.75$, and $a/h \leq 0.8$.

### 6.1. Design Equations

The FEA results of parametric analysis were used to propose design equations for CFS channels with unfastened and fastened flanges when loaded with IOF loading. The proposed equations included the variables, such as $a/h$, $x/h$, and $N/h$. Regression analysis was performed to develop these equations (Equations (2) and (3)) as shown below:

For CFS sections with centered holes:

$$R_{prop} = \alpha' - \gamma'\frac{a}{h} + \lambda'\frac{N}{h} \leq 1 \qquad (2)$$

For CFS sections with offset holes:

$$R_{prop} = \beta' - \mu'\frac{a}{h} + \zeta'\frac{N}{h} + \xi'\frac{x}{h} \leq 1 \qquad (3)$$

where, $\alpha'$, $\gamma'$, $\lambda'$, $\beta'$, $\mu'$, $\zeta'$, and $\xi'$ are the equation coefficients. The equation coefficient values for cold-formed steel channels are summarized in Table 5. The effect of $a/h$, $x/h$, and $N/h$ on the reduced web crippling strength is considered in the equations.

**Table 5.** Proposed equations summary for the web crippling strength reduction factor.

| Coefficients | IOF Loading Condition | |
|:---:|:---:|:---:|
| | Flange Unfastened to Support | Flange Fastened to Support |
| $\alpha'$ | 1.128 | 1.214 |
| $\gamma'$ | 0.378 | 0.537 |
| $\lambda'$ | 0.010 | 0.010 |
| $\beta'$ | 0.618 | 0.932 |
| $\mu'$ | 0.060 | 0.062 |
| $\zeta'$ | 0.047 | 0.084 |
| $\xi'$ | 0.413 | 0.010 |

Table 5 shows that the results obtained from the proposed reduction factors ($R_{prop}$) could closely predict the web crippling failure load of the CFS sections. From Tables 6 and 7, it can be seen that the ratios for $R/R_{prop}$ range from 1.00 to 1.05 and 0.98 to 1.06, for most of the unfastened sections and fastened sections, respectively. The average values of $R/R_{prop}$ are 1.03, with COVs at 0.04 and 0.06, respectively for unfastened sections and fastened sections. Compared to the ratios calculated by the proposed equations of Lian et al. [29,30] ($R/R_{Ying}$) and AISI [48] and AS/NZS [49] ($R/R_{AISI\&AS/NZS}$), the average values of $R/R_{prop}$ are lower with the lower coefficient of variations (COVs). The comparison shows that the proposed equations perform better than those from the other methods in predicting the IOF strength reduction factor ($R$) for both the case of ambient and elevated temperatures.

*6.2. Reliability Analysis*

A detailed reliability analysis was carried out using the methods outlined by Hsiao et al. [57] and Fang et al. [58–60]. In accordance with the American standard [48], when the reliability index of any equation is higher than or equal to the target reliability index 2.5, the equation can be considered reliable:

$$\beta = \frac{\ln(R_m/Q_m)}{\sqrt{V_R^2 + V_Q^2}} \tag{4}$$

where,

$$R_m = R_n M_m F_m P_m \tag{5}$$

$$Q_m = C(D_m + L_m) \tag{6}$$

$$V_R = \sqrt{V_M^2 + V_F^2 + V_P^2} \tag{7}$$

$$V_Q = \frac{\sqrt{D_m^2 V_D^2 + L_m^2 V_L^2}}{D_m + L_m} \tag{8}$$

Here, $R_n$ is the nominal resistance, $M_m$, $F_m$, and $P_m$ are the mean values of the dimensionless random variables reflecting the uncertainties in the material properties, the geometry of the cross section, and the prediction of the ultimate resistance, respectively. $V_R$ and $V_Q$ are the corresponding coefficients of variation. $C$ is a deterministic influence coefficient. $D_m$ and $L_m$ are the mean values. $V_D$ and $V_L$ are the coefficients of variation of the dead load and live load, respectively.

As shown in Tables 8 and 9, the reliability index (2.53 and 2.70 for unfastened sections with centered and offset web holes, respectively; 2.71 and 2.59 for fastened sections with centered and offset web holes, respectively) determined for the proposed equations are all greater than the target reliability index of 2.5 as per the American standard [48] for CFS channels with unfastened and fastened flanges. This shows that the proposed equations are reliable when predicting the IOF web crippling strength of CFS channels with web holes at elevated temperatures. The reliability index for the equations proposed by Lian et al. [29,30] and AISI [48] and AS/NZS [49] were summarized in Tables 8 and 9. In addition, most of the calculated reliability index values of these two methods [29,30,48,49] are lower than the target index value (2.5). Furthermore, a comparison of the reliability index determined for the proposed equations with the reliability index of equations proposed by Lian et al. [29,30] and AISI [48] and AS/NZS [49] was conducted, showing that the proposed equations are more reliable than the equations from Lian et al. [29,30] and AISI [48] and AS/NZS [49], which is in line with the conclusion made in Section 6.1 of the paper.

**Table 6.** Comparison of proposed equations with other methods for sections with unfastened flanges.

| Specimen | Failure Load $P_{A0}$ (kN) | Reduction Factor $R = P_w/P_{A0}$ | | Reduction Factor $R_{AISI\&AS/NZS}$ by AISI&AS/NZS | | Reduction Factor $R_{Ying}$ by Ying et al. | | Reduction Factor $R_{prop}$ by Equation | | $R/R_{AISI\&AS/NZS}$ | | $R/R_{Ying}$ | | $R/R_{prop}$ | |
|---|---|---|---|---|---|---|---|---|---|---|---|---|---|---|---|
| | No Hole | Centered Hole | Offset Hole | Centered Hole | Offset Hole | Centered Hole | Offset Hole | Centered Hole | Offset Hole | Centered Hole | Offset Hole | Centered Hole | Offset Hole | Centered Hole | Offset Hole |
| CFS channel sections at 20 °C | | | | | | | | | | | | | | | |
| 200 × 60 × 30-t1.5-N100-A0.2-FR | 20.01 | 0.99 | 0.97 | 0.93 | 0.93 | 0.96 | - | - | 0.95 | 0.94 | 0.96 | 0.97 | - | - | 0.98 |
| 200 × 60 × 30-t1.5-N100-A0.4-FR | 20.01 | 0.96 | 0.93 | 0.92 | 0.92 | 0.91 | 0.96 | 0.98 | 0.90 | 0.96 | 0.98 | 0.95 | 1.03 | 1.02 | 0.96 |
| 200 × 60 × 30-t1.5-N100-A0.6-FR | 20.01 | 0.88 | 0.86 | 0.90 | 0.90 | 0.86 | 0.90 | 0.91 | 0.85 | 1.03 | 1.05 | 0.97 | 1.04 | 1.03 | 0.98 |
| 200 × 60 × 30-t1.5-N100-A0.8-FR | 20.01 | 0.75 | 0.75 | 0.89 | 0.89 | 0.80 | 0.83 | 0.83 | 0.79 | 1.18 | 1.19 | 1.07 | 1.12 | 1.10 | 1.06 |
| Average | | | | | | | | | | 1.03 | 1.04 | 0.99 | 1.06 | 1.05 | 1.00 |
| COV | | | | | | | | | | 0.09 | 0.09 | 0.05 | 0.04 | 0.04 | 0.04 |
| CFS channel sections at 100 °C | | | | | | | | | | | | | | | |
| 200 × 60 × 30-t1.5-N100-A0.2-FR | 20.01 | 0.98 | 0.97 | 0.93 | 0.93 | 0.96 | - | - | 0.95 | 0.95 | 0.96 | 0.98 | - | - | 0.98 |
| 200 × 60 × 30-t1.5-N100-A0.4-FR | 20.01 | 0.95 | 0.93 | 0.92 | 0.92 | 0.91 | 0.96 | 0.98 | 0.90 | 0.97 | 0.98 | 0.96 | 1.03 | 1.04 | 0.97 |
| 200 × 60 × 30-t1.5-N100-A0.6-FR | 20.01 | 0.87 | 0.86 | 0.90 | 0.90 | 0.86 | 0.90 | 0.91 | 0.85 | 1.04 | 1.05 | 0.99 | 1.05 | 1.05 | 0.99 |
| 200 × 60 × 30-t1.5-N100-A0.8-FR | 20.01 | 0.80 | 0.74 | 0.89 | 0.89 | 0.80 | 0.83 | 0.83 | 0.79 | 1.12 | 1.20 | 1.01 | 1.12 | 1.04 | 1.07 |
| Average | | | | | | | | | | 1.02 | 1.05 | 0.98 | 1.07 | 1.04 | 1.00 |
| COV | | | | | | | | | | 0.07 | 0.09 | 0.02 | 0.04 | 0.00 | 0.04 |
| CFS channel sections at 200 °C | | | | | | | | | | | | | | | |
| 200 × 60 × 30-t1.5-N100-A0.2-FR | 20.01 | 0.95 | 0.97 | 0.93 | 0.93 | 0.96 | - | - | 0.95 | 0.98 | 0.96 | 1.01 | - | - | 0.98 |
| 200 × 60 × 30-t1.5-N100-A0.4-FR | 20.01 | 0.91 | 0.91 | 0.92 | 0.92 | 0.91 | 0.96 | 0.98 | 0.90 | 1.01 | 1.00 | 1.00 | 1.05 | 1.08 | 0.98 |
| 200 × 60 × 30-t1.5-N100-A0.6-FR | 20.01 | 0.83 | 0.84 | 0.90 | 0.90 | 0.86 | 0.90 | 0.91 | 0.85 | 1.08 | 1.08 | 1.03 | 1.07 | 1.09 | 1.01 |
| 200 × 60 × 30-t1.5-N100-A0.8-FR | 20.01 | 0.78 | 0.72 | 0.89 | 0.89 | 0.80 | 0.83 | 0.83 | 0.79 | 1.14 | 1.23 | 1.03 | 1.16 | 1.06 | 1.10 |
| Average | | | | | | | | | | 1.05 | 1.07 | 1.02 | 1.09 | 1.08 | 1.02 |
| COV | | | | | | | | | | 0.06 | 0.10 | 0.01 | 0.05 | 0.01 | 0.05 |
| CFS channel sections at 300 °C | | | | | | | | | | | | | | | |
| 200 × 60 × 30-t1.5-N100-A0.2-FR | 20.01 | 0.94 | 0.97 | 0.93 | 0.93 | 0.96 | - | - | 0.95 | 0.99 | 0.96 | 1.02 | - | - | 0.98 |
| 200 × 60 × 30-t1.5-N100-A0.4-FR | 20.01 | 0.93 | 0.91 | 0.92 | 0.92 | 0.91 | 0.96 | 0.98 | 0.90 | 0.99 | 1.01 | 0.98 | 1.06 | 1.06 | 0.99 |
| 200 × 60 × 30-t1.5-N100-A0.6-FR | 20.01 | 0.83 | 0.83 | 0.90 | 0.90 | 0.86 | 0.90 | 0.91 | 0.85 | 1.09 | 1.09 | 1.03 | 1.09 | 1.09 | 1.02 |

**Table 6.** *Cont.*

| Specimen | Failure Load $P_{A0}$ (kN) No Hole | Reduction Factor $R = P_w/P_{A0}$ Centered Hole | Offset Hole | Reduction Factor $R_{AISI\&AS/NZS}$ by AISI&AS/NZS Centered Hole | Offset Hole | Reduction Factor $R_{Ying}$ by Ying et al. Centered Hole | Offset Hole | Reduction Factor $R_{prop}$ by Equation Centered Hole | Offset Hole | $R/R_{AISI\&AS/NZS}$ Centered Hole | Offset Hole | $R/R_{Ying}$ Centered Hole | Offset Hole | $R/R_{prop}$ Centered Hole | Offset Hole |
|---|---|---|---|---|---|---|---|---|---|---|---|---|---|---|---|
| 200 × 60 × 30-t1.5-N100-A0.8-FR | 20.01 | 0.78 | 0.72 | 0.89 | 0.89 | 0.80 | 0.83 | 0.83 | 0.79 | 1.14 | 1.23 | 1.03 | 1.16 | 1.06 | 1.10 |
| Average | | | | | | | | | | 1.05 | 1.07 | 1.01 | 1.10 | 1.07 | 1.02 |
| COV | | | | | | | | | | 0.07 | 0.10 | 0.02 | 0.04 | 0.02 | 0.05 |
| CFS channel sections at 400 °C | | | | | | | | | | | | | | | |
| 200 × 60 × 30-t1.5-N100-A0.2-FR | 20.01 | 0.95 | 0.97 | 0.93 | 0.93 | 0.96 | - | - | 0.95 | 0.98 | 0.96 | 1.01 | - | - | 0.98 |
| 200 × 60 × 30-t1.5-N100-A0.4-FR | 20.01 | 0.91 | 0.91 | 0.92 | 0.92 | 0.91 | 0.96 | 0.98 | 0.90 | 1.00 | 1.01 | 0.99 | 1.06 | 1.07 | 0.99 |
| 200 × 60 × 30-t1.5-N100-A0.6-FR | 20.01 | 0.84 | 0.83 | 0.90 | 0.90 | 0.86 | 0.90 | 0.91 | 0.85 | 1.07 | 1.09 | 1.02 | 1.09 | 1.08 | 1.02 |
| 200 × 60 × 30-t1.5-N100-A0.8-FR | 20.01 | 0.80 | 0.73 | 0.89 | 0.89 | 0.80 | 0.83 | 0.83 | 0.79 | 1.11 | 1.22 | 1.01 | 1.15 | 1.04 | 1.09 |
| Average | | | | | | | | | | 1.04 | 1.07 | 1.01 | 1.10 | 1.06 | 1.02 |
| COV | | | | | | | | | | 0.05 | 0.10 | 0.01 | 0.04 | 0.02 | 0.04 |
| CFS channel sections at 500 °C | | | | | | | | | | | | | | | |
| 200 × 60 × 30-t1.5-N100-A0.2-FR | 20.01 | 0.99 | 0.97 | 0.93 | 0.93 | 0.96 | - | - | 0.95 | 0.94 | 0.96 | 0.97 | - | - | 0.98 |
| 200 × 60 × 30-t1.5-N100-A0.4-FR | 20.01 | 0.97 | 0.93 | 0.92 | 0.92 | 0.91 | 0.96 | 0.98 | 0.90 | 0.95 | 0.99 | 0.94 | 1.04 | 1.01 | 0.97 |
| 200 × 60 × 30-t1.5-N100-A0.6-FR | 20.01 | 0.89 | 0.85 | 0.90 | 0.90 | 0.86 | 0.90 | 0.91 | 0.85 | 1.01 | 1.07 | 0.96 | 1.06 | 1.02 | 1.00 |
| 200 × 60 × 30-t1.5-N100-A0.8-FR | 20.01 | 0.76 | 0.73 | 0.89 | 0.89 | 0.80 | 0.83 | 0.83 | 0.79 | 1.17 | 1.22 | 1.06 | 1.14 | 1.10 | 1.09 |
| Average | | | | | | | | | | 1.02 | 1.06 | 0.98 | 1.08 | 1.04 | 1.01 |
| COV | | | | | | | | | | 0.09 | 0.10 | 0.05 | 0.05 | 0.04 | 0.05 |
| CFS channel sections at 600 °C | | | | | | | | | | | | | | | |
| 200 × 60 × 30-t1.5-N100-A0.2-FR | 20.01 | 0.98 | 0.96 | 0.93 | 0.93 | 0.96 | - | - | 0.95 | 0.95 | 0.97 | 0.97 | - | - | 0.99 |
| 200 × 60 × 30-t1.5-N100-A0.4-FR | 20.01 | 0.94 | 0.90 | 0.92 | 0.92 | 0.91 | 0.96 | 0.98 | 0.90 | 0.97 | 1.02 | 0.96 | 1.07 | 1.04 | 1.00 |
| 200 × 60 × 30-t1.5-N100-A0.6-FR | 20.01 | 0.88 | 0.81 | 0.90 | 0.90 | 0.86 | 0.90 | 0.91 | 0.85 | 1.03 | 1.12 | 0.97 | 1.11 | 1.03 | 1.05 |
| 200 × 60 × 30-t1.5-N100-A0.8-FR | 20.01 | 0.82 | 0.68 | 0.89 | 0.89 | 0.80 | 0.83 | 0.83 | 0.79 | 1.08 | 1.30 | 0.98 | 1.22 | 1.01 | 1.16 |
| Average | | | | | | | | | | 1.01 | 1.10 | 0.97 | 1.13 | 1.03 | 1.05 |

**Table 6.** *Cont.*

| Specimen | Failure Load P_A0 (kN) no Hole | Reduction Factor R = Pw/PA0 | | Reduction Factor R_AISI&AS/NZS by AISI&AS/NZS | | Reduction Factor R_Ying by Ying et al. | | Reduction Factor R_prop by Equation | | R/R_AISI&AS/NZS | | R/R_Ying | | R/R_prop | |
|---|---|---|---|---|---|---|---|---|---|---|---|---|---|---|---|
| | | Centered Hole | Offset Hole | Centered Hole | Offset Hole | Centered Hole | Offset Hole | Centered Hole | Offset Hole | Centered Hole | Offset Hole | Centered Hole | Offset Hole | Centered Hole | Offset Hole |
| COV | | | | | | | | | | 0.05 | 0.13 | 0.01 | 0.07 | 0.01 | 0.07 |
| CFS channel sections at 700 °C | | | | | | | | | | | | | | | |
| 200 × 60 × 30-t1.5-N100-A0.2-FR | 20.01 | 0.99 | 0.96 | 0.93 | 0.93 | 0.96 | - | - | 0.95 | 0.94 | 0.97 | 0.97 | - | - | 0.99 |
| 200 × 60 × 30-t1.5-N100-A0.4-FR | 20.01 | 0.97 | 0.92 | 0.92 | 0.92 | 0.91 | 0.96 | 0.98 | 0.90 | 0.94 | 1.00 | 0.93 | 1.05 | 1.01 | 0.98 |
| 200 × 60 × 30-t1.5-N100-A0.6-FR | 20.01 | 0.91 | 0.81 | 0.90 | 0.90 | 0.86 | 0.90 | 0.91 | 0.85 | 0.99 | 1.11 | 0.94 | 1.10 | 1.00 | 1.04 |
| 200 × 60 × 30-t1.5-N100-A0.8-FR | 20.01 | 0.78 | 0.66 | 0.89 | 0.89 | 0.80 | 0.83 | 0.83 | 0.79 | 1.14 | 1.34 | 1.03 | 1.26 | 1.07 | 1.19 |
| Average | | | | | | | | | | 1.00 | 1.10 | 0.97 | 1.14 | 1.03 | 1.05 |
| COV | | | | | | | | | | 0.08 | 0.14 | 0.04 | 0.09 | 0.03 | 0.09 |

**Table 7.** Comparison of proposed equations with other methods for sections with fastened flanges.

| Specimen | Failure Load P_A0 (kN) No Hole | Reduction Factor R = Pw/PA0 | | Reduction Factor R_AISI&AS/NZS by AISI&AS/NZS | | Reduction Factor R_Ying by Ying et al. | | Reduction Factor R_prop by Equation | | R/R_AISI&AS/NZS | | R/R_Ying | | R/R_prop | |
|---|---|---|---|---|---|---|---|---|---|---|---|---|---|---|---|
| | | Centered Hole | Offset Hole | Centered Hole | Offset Hole | Centered Hole | Offset Hole | Centered Hole | Offset Hole | Centered Hole | Offset Hole | Centered Hole | Offset Hole | Centered Hole | Offset Hole |
| CFS channel sections at 20 °C | | | | | | | | | | | | | | | |
| 200 × 60 × 30-t1.5-N100-A0.2-FX | 20.01 | 0.97 | 0.97 | 0.93 | 0.93 | 0.94 | - | - | 0.97 | 0.96 | 0.96 | 0.97 | - | - | 1.00 |
| 200 × 60 × 30-t1.5-N100-A0.4-FX | 20.01 | 0.96 | 0.96 | 0.92 | 0.92 | 0.93 | 0.98 | 1.00 | 0.96 | 0.96 | 0.96 | 0.97 | 1.02 | 1.05 | 1.00 |
| 200 × 60 × 30-t1.5-N100-A0.6-FX | 20.01 | 0.93 | 0.94 | 0.90 | 0.90 | 0.92 | 0.95 | 0.90 | 0.94 | 0.97 | 0.96 | 0.99 | 1.01 | 0.96 | 1.01 |
| 200 × 60 × 30-t1.5-N100-A0.8-FX | 20.01 | 0.74 | 0.86 | 0.89 | 0.89 | 0.91 | 0.91 | 0.79 | 0.93 | 1.21 | 1.03 | 1.23 | 1.06 | 1.07 | 1.08 |
| Average | | | | | | | | | | 1.02 | 0.98 | 1.04 | 1.03 | 1.03 | 1.02 |
| COV | | | | | | | | | | 0.11 | 0.03 | 0.11 | 0.02 | 0.05 | 0.04 |
| CFS channel sections at 100 °C | | | | | | | | | | | | | | | |
| 200 × 60 × 30-t1.5-N100-A0.2-FX | 20.01 | 0.97 | 0.99 | 0.93 | 0.93 | 0.94 | - | - | 0.97 | 0.96 | 0.94 | 0.97 | - | - | 0.98 |

Table 7. *Cont.*

| Specimen | Failure Load $P_{A0}$ (kN) | Reduction Factor $R = P_w/P_{A0}$ | | Reduction Factor $R_{AISI\&AS/NZS}$ by AISI&AS/NZS | | Reduction Factor $R_{Ying}$ by Ying et al. | | Reduction Factor $R_{prop}$ by Equation | | $R/R_{AISI\&AS/NZS}$ | | $R/R_{Ying}$ | | $R/R_{prop}$ | |
|---|---|---|---|---|---|---|---|---|---|---|---|---|---|---|---|
| | No Hole | Centered Hole | Offset Hole | Centered Hole | Offset Hole | Centered Hole | Offset Hole | Centered Hole | Offset Hole | Centered Hole | Offset Hole | Centered Hole | Offset Hole | Centered Hole | Offset Hole |
| 200 × 60 × 30-t1.5-N100-A0.4-FX | 20.01 | 0.96 | 0.96 | 0.92 | 0.92 | 0.93 | 0.98 | 1.00 | 0.96 | 0.96 | 0.96 | 0.97 | 1.02 | 1.05 | 1.00 |
| 200 × 60 × 30-t1.5-N100-A0.6-FX | 20.01 | 0.94 | 0.94 | 0.90 | 0.90 | 0.92 | 0.95 | 0.90 | 0.94 | 0.96 | 0.96 | 0.98 | 1.01 | 0.96 | 1.01 |
| 200 × 60 × 30-t1.5-N100-A0.8-FX | 20.01 | 0.73 | 0.84 | 0.89 | 0.89 | 0.91 | 0.91 | 0.79 | 0.93 | 1.21 | 1.06 | 1.24 | 1.09 | 1.08 | 1.11 |
| Average | | | | | | | | | | 1.02 | 0.98 | 1.04 | 1.04 | 1.03 | 1.02 |
| COV | | | | | | | | | | 0.11 | 0.05 | 0.11 | 0.03 | 0.05 | 0.05 |
| CFS channel sections at 200 °C | | | | | | | | | | | | | | | |
| 200 × 60 × 30-t1.5-N100-A0.2-FX | 20.01 | 0.97 | 0.98 | 0.93 | 0.93 | 0.94 | - | - | 0.97 | 0.96 | 0.95 | 0.97 | - | - | 0.99 |
| 200 × 60 × 30-t1.5-N100-A0.4-FX | 20.01 | 0.96 | 0.96 | 0.92 | 0.92 | 0.93 | 0.98 | 1.00 | 0.96 | 0.96 | 0.96 | 0.97 | 1.02 | 1.05 | 1.00 |
| 200 × 60 × 30-t1.5-N100-A0.6-FX | 20.01 | 0.88 | 0.93 | 0.90 | 0.90 | 0.92 | 0.95 | 0.90 | 0.94 | 1.03 | 0.97 | 1.05 | 1.02 | 1.02 | 1.02 |
| 200 × 60 × 30-t1.5-N100-A0.8-FX | 20.01 | 0.68 | 0.80 | 0.89 | 0.89 | 0.91 | 0.91 | 0.79 | 0.93 | 1.31 | 1.11 | 1.34 | 1.14 | 1.16 | 1.16 |
| Average | | | | | | | | | | 1.06 | 1.00 | 1.08 | 1.06 | 1.08 | 1.04 |
| COV | | | | | | | | | | 0.14 | 0.07 | 0.15 | 0.06 | 0.06 | 0.07 |
| CFS channel sections at 300 °C | | | | | | | | | | | | | | | |
| 200 × 60 × 30-t1.5-N100-A0.2-FX | 20.01 | 0.97 | 0.99 | 0.93 | 0.93 | 0.94 | - | - | 0.97 | 0.96 | 0.94 | 0.97 | - | - | 0.98 |
| 200 × 60 × 30-t1.5-N100-A0.4-FX | 20.01 | 0.96 | 0.96 | 0.92 | 0.92 | 0.93 | 0.98 | 1.00 | 0.96 | 0.96 | 0.95 | 0.97 | 1.02 | 1.05 | 1.00 |
| 200 × 60 × 30-t1.5-N100-A0.6-FX | 20.01 | 0.85 | 0.93 | 0.90 | 0.90 | 0.92 | 0.95 | 0.90 | 0.94 | 1.06 | 0.97 | 1.08 | 1.02 | 1.05 | 1.02 |
| 200 × 60 × 30-t1.5-N100-A0.8-FX | 20.01 | 0.66 | 0.78 | 0.89 | 0.89 | 0.91 | 0.91 | 0.79 | 0.93 | 1.35 | 1.14 | 1.38 | 1.17 | 1.20 | 1.20 |
| Average | | | | | | | | | | 1.08 | 1.00 | 1.10 | 1.07 | 1.10 | 1.05 |
| COV | | | | | | | | | | 0.16 | 0.08 | 0.17 | 0.07 | 0.07 | 0.09 |
| CFS channel sections at 400 °C | | | | | | | | | | | | | | | |
| 200 × 60 × 30-t1.5-N100-A0.2-FX | 20.01 | 0.97 | 0.98 | 0.93 | 0.93 | 0.94 | - | - | 0.97 | 0.96 | 0.95 | 0.97 | - | - | 0.99 |
| 200 × 60 × 30-t1.5-N100-A0.4-FX | 20.01 | 0.96 | 0.96 | 0.92 | 0.92 | 0.93 | 0.98 | 1.00 | 0.96 | 0.96 | 0.95 | 0.97 | 1.02 | 1.05 | 1.00 |
| 200 × 60 × 30-t1.5-N100-A0.6-FX | 20.01 | 0.87 | 0.93 | 0.90 | 0.90 | 0.92 | 0.95 | 0.90 | 0.94 | 1.04 | 0.97 | 1.06 | 1.02 | 1.03 | 1.01 |
| 200 × 60 × 30-t1.5-N100-A0.8-FX | 20.01 | 0.68 | 0.79 | 0.89 | 0.89 | 0.91 | 0.91 | 0.79 | 0.93 | 1.31 | 1.13 | 1.34 | 1.16 | 1.17 | 1.19 |
| Average | | | | | | | | | | 1.07 | 1.00 | 1.09 | 1.07 | 1.08 | 1.05 |

**Table 7.** *Cont.*

| Specimen | Failure Load $P_{A0}$ (kN) | Reduction Factor $R = P_w/P_{A0}$ | | Reduction Factor $R_{AISI\&AS/NZS}$ by AISI&AS/NZS | | Reduction Factor $R_{Ying}$ by Ying et al. | | Reduction Factor $R_{prop}$ by Equation | | $R/R_{AISI\&AS/NZS}$ | | $R/R_{Ying}$ | | $R/R_{prop}$ | |
|---|---|---|---|---|---|---|---|---|---|---|---|---|---|---|---|
| | No Hole | Centered Hole | Offset Hole | Centered Hole | Offset Hole | Centered Hole | Offset Hole | Centered Hole | Offset Hole | Centered Hole | Offset Hole | Centered Hole | Offset Hole | Centered Hole | Offset Hole |
| COV | | | | | | | | | | 0.15 | 0.08 | 0.15 | 0.07 | 0.06 | 0.08 |
| *CFS channel sections at 500 °C* | | | | | | | | | | | | | | | |
| 200 × 60 × 30-t1.5-N100-A0.2-FX | 20.01 | 0.97 | 0.98 | 0.93 | 0.93 | 0.94 | - | - | 0.98 | 0.96 | 0.95 | 0.97 | - | - | 0.99 |
| 200 × 60 × 30-t1.5-N100-A0.4-FX | 20.01 | 0.96 | 0.96 | 0.92 | 0.92 | 0.93 | 0.98 | 1.00 | 0.96 | 0.96 | 0.95 | 0.97 | 1.02 | 1.05 | 1.00 |
| 200 × 60 × 30-t1.5-N100-A0.6-FX | 20.01 | 0.91 | 0.94 | 0.90 | 0.90 | 0.92 | 0.95 | 0.90 | 0.94 | 0.99 | 0.96 | 1.01 | 1.01 | 0.99 | 1.01 |
| 200 × 60 × 30-t1.5-N100-A0.8-FX | 20.01 | 0.73 | 0.84 | 0.89 | 0.89 | 0.91 | 0.91 | 0.79 | 0.93 | 1.22 | 1.06 | 1.25 | 1.09 | 1.09 | 1.11 |
| Average | | | | | | | | | | 1.03 | 0.98 | 1.05 | 1.04 | 1.04 | 1.03 |
| COV | | | | | | | | | | 0.11 | 0.05 | 0.12 | 0.03 | 0.04 | 0.05 |
| *CFS channel sections at 600 °C* | | | | | | | | | | | | | | | |
| 200 × 60 × 30-t1.5-N100-A0.2-FX | 20.01 | 0.97 | 0.99 | 0.93 | 0.93 | 0.94 | - | - | 0.98 | 0.96 | 0.94 | 0.97 | - | - | 0.99 |
| 200 × 60 × 30-t1.5-N100-A0.4-FX | 20.01 | 0.96 | 0.96 | 0.92 | 0.92 | 0.93 | 0.98 | 1.00 | 0.96 | 0.96 | 0.95 | 0.97 | 1.02 | 1.05 | 0.99 |
| 200 × 60 × 30-t1.5-N100-A0.6-FX | 20.01 | 0.95 | 0.95 | 0.90 | 0.90 | 0.92 | 0.95 | 0.90 | 0.94 | 0.95 | 0.95 | 0.97 | 1.00 | 0.94 | 1.00 |
| 200 × 60 × 30-t1.5-N100-A0.8-FX | 20.01 | 0.80 | 0.91 | 0.89 | 0.89 | 0.91 | 0.91 | 0.79 | 0.93 | 1.12 | 0.98 | 1.14 | 1.00 | 0.99 | 1.03 |
| Average | | | | | | | | | | 0.99 | 0.96 | 1.01 | 1.01 | 0.99 | 1.00 |
| COV | | | | | | | | | | 0.07 | 0.01 | 0.07 | 0.01 | 0.04 | 0.02 |
| *CFS channel sections at 700 °C* | | | | | | | | | | | | | | | |
| 200 × 60 × 30-t1.5-N100-A0.2-FX | 20.01 | 0.97 | 0.99 | 0.93 | 0.93 | 0.94 | - | - | 0.98 | 0.96 | 0.94 | 0.97 | - | - | 0.99 |
| 200 × 60 × 30-t1.5-N100-A0.4-FX | 20.01 | 0.96 | 0.97 | 0.92 | 0.92 | 0.93 | 0.98 | 0.97 | 0.96 | 0.96 | 0.94 | 0.97 | 1.01 | 1.01 | 0.99 |
| 200 × 60 × 30-t1.5-N100-A0.6-FX | 20.01 | 0.94 | 0.97 | 0.90 | 0.90 | 0.92 | 0.95 | 0.89 | 0.95 | 0.96 | 0.93 | 0.98 | 0.98 | 0.94 | 0.98 |
| 200 × 60 × 30-t1.5-N100-A0.8-FX | 20.01 | 0.82 | 0.96 | 0.89 | 0.89 | 0.91 | 0.91 | 0.80 | 0.94 | 1.08 | 0.92 | 1.11 | 0.94 | 0.97 | 0.97 |
| Average | | | | | | | | | | 0.99 | 0.93 | 1.01 | 0.98 | 0.98 | 0.98 |
| COV | | | | | | | | | | 0.05 | 0.01 | 0.06 | 0.03 | 0.03 | 0.01 |

**Table 8.** Reliability analysis results (centered-hole sections).

|  | **With Fastened Flanges** | **With Unfastened Flanges** |
|---|---|---|
| Ratio of equations | $R_{FEA}/R_{prop}$ | $R_{FEA}/R_{prop}$ |
| Data number | 381 | 415 |
| Mean, $P_m$ | 1.00 | 1.00 |
| Reliability index, $\beta$ | 2.71 | 2.53 |
| Reliability index, $\beta$ for AISI&AS/NZS | 2.39 | 2.23 |
| Reliability index, $\beta$ for Ying et al. | 2.70 | 2.12 |
| Reliability index, $\varphi$ | 0.85 | 0.85 |

**Table 9.** Reliability analysis of the proposed equations (offset-hole sections).

|  | **With Fastened Flanges** | **With Unfastened Flanges** |
|---|---|---|
| Ratio of equations | $R_{FEA}/R_{prop}$ | $R_{FEA}/R_{prop}$ |
| Data number | 756 | 748 |
| Mean, $P_m$ | 1.00 | 1.01 |
| Reliability index, $\beta$ | 2.59 | 2.70 |
| Reliability index, $\beta$ for AISI&AS/NZS | 2.47 | 2.69 |
| Reliability index, $\beta$ for Ying et al. | 2.42 | 2.63 |
| Reliability index, $\varphi$ | 0.85 | 0.85 |

## 7. Summary and Conclusions

This study carried out a numerical investigation on the IOF web crippling behavior of CFS channels with web holes at elevated temperatures. The two popular carbon steel grades, i.e., G250 and G450 were used. A nonlinear FE model was developed for CFS perforated channels under IOF loading and was validated against the test results available in the literature for ambient temperatures. Using the validated FE model of perforated CFS channels at ambient temperatures, an extensive parametric analysis was undertaken to investigate the effects of fire loading on its web crippling strength. In total, 3474 FEA models were analyzed. In the parametric study, the effects of $a/h$, $x/h$, and $N/h$ flange type and temperatures on the web crippling strength of CFS channels with web holes at elevated temperatures were discussed. According to the parametric study results, the web crippling strength reduction factor is sensitive to changes in the ratios of $a/h$, $N/h$, and $x/h$, with the ratios of $a/h$ and $x/h$ having the largest effects on the web crippling reduction factor. However, the web crippling strength reduction factor remains stable when the temperature is changed from 20 to 700 °C.

New design equations were developed to consider the effects of cross-sectional geometry, plate element, flange type, hole size, and hole position to obtain the IOF web crippling strength of CFS channels at elevated temperatures. The limits for the proposed equations are $h/t \leq 160$, $N/t \leq 120$, $N/h \leq 0.75$, and $a/h \leq 0.8$. Compared with the proposed equations of Lian et al. (2017), a new factor $N/h$ was included to the proposed design equations on the strength reduction factor, which was ignored by Lian et al. (2017). In addition to the influence of cross-section and hole size parameters, other parameters, such as hole position and flange type also play an important role in the failure strength of the web. Therefore, these effects are included in the proposed equations. The comparison of results obtained from the current design standards and from the proposed equations of the current research and from Lian et al. (2017) showed that the equations developed in this research performed better than the other equations by at least 5% on the web crippling strength of both plain and perforated channels. Thereafter, a comprehensive reliability analysis was performed, which showed that the proposed design equations are capable of demonstrating a reliable limit state design on the web crippling performance of perforated CFS channel sections when calibrated with a resistance factor from the American standard.

Some limitations were set in this study, in terms of applicability of the proposed design equations. First, for the offset-hole sections used in this paper, the proposed equations should be considered for sections with two symmetrically offset web holes. Meanwhile, the hole sizes for each offset web hole should be equal. Second, in this paper, the proposed equations cover the typical IOF web crippling case. However, in real engineering practice, the loading and boundary conditions may vary. Third, only two typical carbon steel material properties were considered in the FE modelling. The structural behavior of channels with other grades of carbon steel should be investigated in the future. Finally, the prediction accuracy of proposed equations was only assessed based on the validated FEA. The experimental tests of CFS at elevated temperatures subjected to web crippling should be conducted in the future.

The limitations as mentioned above show the need of future studies in the following areas:

- For the case of sections with one offset web holes, and sections with two non-symmetrically offset web holes with different hole diameters, the web crippling coefficients of design equations are needed to be developed.
- In the case of sections with complicated loading and boundary conditions, the modification of web crippling coefficients of design standards should be made.
- The structural behavior of perforated channels with other grades of carbon steel should be investigated.
- The experimental tests of CFS channels subjected to web crippling at elevated temperatures should be conducted.

**Author Contributions:** Conceptualization, Z.F. and K.R.; data curation, Z.F. and H.L.; formal analysis, Z.F. and K.G.; methodology, Z.F., K.P. and A.M.M.; supervision, K.R., K.P., A.M.M. and J.B.P.L.; validation, Z.F. and H.L.; writing—original draft, Z.F.; writing—review & editing, Z.F., K.R. and J.B.P.L. All authors have read and agreed to the published version of the manuscript.

**Funding:** This research received no external funding.

**Institutional Review Board Statement:** Not applicable.

**Informed Consent Statement:** Not applicable.

**Data Availability Statement:** Not applicable.

**Acknowledgments:** The authors thank the support given by the University of Auckland for providing them with high-performance calculating and computing machines.

**Conflicts of Interest:** The authors declare no conflict of interest.

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
