# Peer review of "Numerical Simulation and Design Recommendations for Web Crippling Strength of Cold-Formed Steel Channels with Web Holes under Interior-One-Flange Loading at Elevated Temperatures"

_buildings, doi:10.3390/buildings11120666_

Round 1

Reviewer 1 Report

This article presents the research on the interior-one-flange web crippling strength of cold-formed steel channels at elevated temperatures.

FEM analysis for the mechanical properties of steel elements was based on models from experimental studies described in the related literature positions with research prepared for ambient temperatures conditions. In the reviewer's opinion, the proposed models should be confirmed by experimental tests of mechanical properties of the tested elements at elevated temperatures.
Some of the results of the research were limited to the presentation of calculated data in the tables. There is a lack of their relevant analysis, and above all, a discussion of the obtained research results. Reliability analysis should indicate the superiority of the presented calculation formulae in relation to other studies cited in the literature. It should be clearly indicated in the discussion what the novelty and superiority of the presented method is in relation to other calculation methods cited and compared in the article.
There are no conclusions from the conducted research. The information presented in the conclusions chapter is a summary of the presented research work. 

Yours Sincerely,

Reviewer.

Author Response

The authors sincerely thank the Reviewer for her/his positive comments and for the time spent reviewing the paper. We have now replied to your questions one by one and made necessary changes to the manuscript. The point-by-point responses to each of your questions are given in the attached file. 

Reviewer 2 Report

Dear Author(s),

I have reviewed your paper "A Numerical Study and Proposed Design Rules for Web Crippling Strength of Cold-Formed Steel Channels with Web Holes under Interior-One-Flange Loading at Elevated Temperature", which has been submitted to Buildings.

The paper fulfills the aims and scope of the journal. Presented investigations are interesting and worth to be considered for potential publishing. I have some questions/comments, which are listed below.

General remarks:

  • The quantitative results should be listed in the abstract.

Introduction:

  • "Cold-formed steel (CFS) is used increasingly in commercial and residential buildings because of its superior strength to weight ratio, stiffness, and ease of construction [1-18]" - in my opinion, this sentence should be modified. You should described 1-18 references in more details. You should avoid presenting so many references in one brackets.
  • The rest of introcustion is well prepared. You have presented relevant scientific background for your studies.

Summary of Experimental Investigation from Lian et al. [29, 30]:

  • The name of section could be modified. You could delete "frome Lian et al. [29,30] - please present this information in first sentence of the section.

Numerical simulation:

  • Please mark in the text how many elements were used in simulations. Moreover, plase add information about their shape.
  • Table 1 - the last column is cover - the "Pfea" is not visible in full.
  • Table 2 - the same as above.

Current Design Rules:

  • Lines  138-139 - "For IOF loading, where any portion of a web hole is not within the bearing length, the reduction factor, R, can be calculated as follows:". This statement should be supported by references, or relevant description.

Parametric analysis:

  • It is not clear, what do the colour at plots mean (Figs 4 and 5). It should be described in the pictures.
  • You should analyse your results in comparison to other results available in literature. It will underline the advantages of your investigations.

Conclusions:

  • I propose to present the most important results in points. It will be more readable. Moreover, please support results with quantitative results.

Author Response

(The authors gave the same response as above.)

Round 2

Reviewer 1 Report

The data presented in tables 6-9 should be commented on afterwards, f.e. with the information on the lowest and the highest reduction factors achieved in the presented comparisons and other important presented positions.

Yours Sincerely,

Reviewer. 

Author Response

The authors are grateful to the editor and the reviewers for their responses to our paper and are grateful for their positive comments. Some issues have been raised by the two reviewers and these have been addressed. The review comments are reproduced in black in the accompanying pages and a detailed response by the authors to each one is presented in red. All the amendments which have been made to the revised manuscript are highlighted in yellow.
